# THE EXPRESSIVE POWER OF DEEP NEURAL NETWORKS WITH CIRCULANT MATRICES

## ABSTRACT

Recent results from linear algebra stating that any matrix can be decomposed into products of diagonal and circulant matrices has lead to the design of compact deep neural network architectures that perform well in practice. In this paper, we bridge the gap between these good empirical results and the theoretical approximation capabilities of Deep diagonal-circulant ReLU networks. More precisely, we first demonstrate that a Deep diagonal-circulant ReLU networks of bounded width and small depth can approximate a deep ReLU network in which the dense matrices are of low rank. Based on this result, we provide new bounds on the expressive power and universal approximativeness of this type of networks. We support our experimental results with thorough experiments on a large, real world video classification problem.

## 1 INTRODUCTION

Recent progress in deep neural networks came at the cost of an important increase of model sizes. Nowadays, state-of-the-art architectures for common tasks such as object recognition typically have tens of millions of parameters (He et al., 2016) and up to a billion parameters in some cases (Dean et al., 2012). Best performing (ensemble) models typically combine dozens of such models, and their size can quickly add up to ten or twenty gigabytes. Large models are often more accurate, but training them requires time and large amounts of computational resources. Even when they are trained, they remain difficult to deploy, especially on mobile devices where memory or computational power is limited.

In linear algebra, it is common to exploit structural properties of matrices to speedup computations, or reduce memory usage. Cheng et al. (2015) have applied this principle in the context of deep neural networks, and proposed a network architecture in which large unstructured weight matrices have been replaced with more compact matrices with a *circulant* structure. Since any $n$-by-$n$ circulant matrix can be represented in memory using only a vector of dimension $n$, the change resulted in a drastic reduction of the model size (from 230MB to 21MB). Furthermore, Cheng et al. have shown empirically that their network architecture can be almost as accurate as the original network.

Moczulski et al. (2015) have proposed a more principled approach leveraging a result by Huhtanen & Perämäki (2015) stating that any matrix $A \in \mathbb{C}^{n \times n}$ can be decomposed into $2n - 1$ diagonal and circulant matrices. They use this result to design Deep diagonal-circulant ReLU networks. However their experiments show good results even with a small number of factors (down to 2 factors), suggesting that Deep diagonal-circulant ReLU networks can achieve good approximation error, even with few factors.

In this paper, we bridge the gap between the good empirical results observed by Moczulski et al. (2015), and the theoretical approximation capabilities of Deep diagonal-circulant ReLU networks. We prove that Deep diagonal-circulant ReLU networks with bounded width and small depth can approximate any dense neural network. We obtain this result by showing that any matrix $A$ can be decomposed into $4k + 1$ diagonal and

circulant matrices where $k$ is the rank of the matrix $A$. In practice, this result is more useful than the one by Huhtanen & Perämäki since one can rely on a low rank SVD decomposition of $A$ while controlling the approximation error.

In addition to this theoretical contribution, we also conduct thorough experiments on synthetic and real datasets. In accordance with the theory, our experiments demonstrate that we can easily tradeoff accuracy for model size by adjusting the number of factors in the matrix decomposition. Finally we evaluate the applicability of this approach on state-of-the-art neural network architectures trained for video classification on the Youtube-8m Video dataset (over 1TB of training data). This experiment demonstrates that Deep diagonal-circulant ReLU networks can be used to train more compact neural networks on large scale, real world scenarios.

## 2 RELATED WORK

A variety of techniques have been proposed to build more compact deep learning models. A first category of techniques aims at *compressing* a trained network into a smaller model, without compromising the accuracy. For example *model distillation* (Hinton et al., 2015) is two step training procedure: first a large model is trained to be as accurate as possible, second a more compact model is trained to approximate the first one. Other approaches have focused on compressing the network by reducing the memory, at the level of individual weights, (for example, using weight quantization (Han et al., 2016) or parameter pruning) or at the level of weight matrices, using low rank decomposition of the original weight matrix (Sainath et al., 2013) or using sparse representations (Collins & Kohli, 2014; Dai et al., 2018; Liu et al., 2015).

Instead of compressing the network *a posteriori*, several researchers have focused on designing models that are compact by design. This approach has several benefits, but most importantly, it reduces memory footprint, both required during training and inference. Chen et al. (2015) have proposed to compress weight matrices by using hashing functions to map several matrix coefficients into the same memory cell. The techniques works well in theory, but suffers from poor performance on modern GPU devices due to irregular access patterns.

In their paper, Cheng et al. (2015) observed that fully connected layers (which typically occupy 90%[1]. of the total number of weights) are often used to perform simple dimensionality reduction operation between layers of different dimension. The idea of replacing large weight matrices from fully connected layers with more compact circulant layers comes from a result by Hinrichs & Vybíral (2011) that have demonstrated that circulant matrices can be use to approximate the Johson-Lindenstrauss transform, often used to perform dimensionality reduction. Building on this result Cheng et al. proposed to replace the weight matrix of a fully connected layer by a circulant matrix initialized with random weights. The resulting models achieve good accuracy, with the random circulant matrix, but even better when the weights of the circulant matrix are trained with the rest of the network using a gradient based optimization algorithm. This suggests that such layers, often perform more than simple random projections, and that more expressive fully connected layers are beneficial to the overall accuracy of the model.

Fortunately, more general linear transforms can also be described using circulant matrices or other structured matrices, at the cost of using more of them. Müller-Quade et al. (1998) and Schmid et al. (2000) have demonstrated this formally by showing that *any* matrix can be decomposed into the product of diagonal and circulant matrices, and Moczulski et al. (2015) have proposed a compact neural network architecture based on this decomposition that exhibit good accuracy in practice. Other researchers have investigated using alternative structures such as Toeplitz (Sindhwani et al., 2015), Vandermonde (Sindhwani et al., 2015) or Fastfood transforms (Yang et al., 2015). Despite demonstrating good empirical results, there have been

---

[1]In network such as AlexNet, the last 3 fully connected layers use 58M out of the 62M total trainable parameters.

little theoretical insight to explain the good approximation capabilities of deep neural networks based on structured matrices.

Barron (1993) presented the *universal approximation theorem* which states that any neural network with at least 1 hidden layer and sigmoid non linearity can approximate any function. However, the theorem by Barron (1993) does not bound the width of the neural network and does not consider the training procedure. Since then, substantial theoretical work has been done to evaluate the expressiveness of a neural network as a function of the width (i.e. the number of neurons) and the depth of the network Arora et al. (2018); Mhaskar et al. (2017); Lin et al. (2017); Poole et al. (2016); Raghu et al. (2016); Telgarsky (2016); Mhaskar & Poggio (2016). In 2000, Hanin (2017) have investigated the approximation capabilities of neural networks with ReLU activations and demonstrated that such networks can approximate any function.

More recently, Zhao et al. (2017) have provided a theoretical study of Deep diagonal-circulant ReLU networks and demonstrated that 2-layers networks of unbounded width are universal approximators. However, these results are of limited interest because the networks used in practice are of bounded width. Unfortunately, nothing is known about the theoretical properties of Deep diagonal-circulant ReLU networks in this case.

## 3 BUILDING COMPACT DEEP NEURAL NETWORKS USING CIRCULANT MATRICES

### 3.1 PRELIMINARIES ON CIRCULANT MATRICES

A n-by-n circulant matrix $C$ is a special kind of Toeplitz matrix where each row is a cyclic right shift of the previous one as illustrated below.

$$C = circ(c) = \begin{bmatrix} c_0 & c_{n-1} & c_{n-2} & \dots & c_1 \\ c_1 & c_0 & c_{n-1} & & c_2 \\ c_2 & c_1 & c_0 & & c_3 \\ \vdots & & & \ddots & \vdots \\ c_{n-1} & c_{n-2} & c_{n-3} & & c_0 \end{bmatrix}$$

Despite their rigorous structure, circulant matrices are expressive enough to model a variety of linear transforms such as random projections (Hinrichs & Vybíral, 2011) and when they are combined together with diagonal matrices, they can be used to represent an arbitrary transform (Schmid et al., 2000).

Circulant matrices also exhibit several properties that are interesting from a computational perspective. First, a circulant $n$-by-$n$ matrix $C$ can be represented using only $n$ coefficients. Thus, it is far more compact that a full matrix that requires $n^2$ coefficient. Second, the product between a circulant matrix $C$ and a vector $x$ can be simplified to a simple element-wise product between the vector $c$ and $x$ in the Fourier domain (which is generally performed efficiently on GPU devices). This results in a complexity reduced from $O(n^2)$ to $O(nlog(n))$.

In their paper, Huhtanen & Perämäki (2015) have demonstrated that any matrix $A \in \mathbb{C}^{n \times n}$ can be approximated with an arbitrary precision by a product of circulant and diagonal matrices:

**Theorem 1.** *(Huhtanen & Perämäki, 2015) For any given matrix $A \in \mathbb{C}^{n \times n}$, let $p$ be the smallest integer such that $A = \sum_{i=1}^{p} D_i S^{i-1}$ where $D_1 \dots D_p$ are diagonal matrices. Then for any $\epsilon > 0$, for any matrix norm $\|\cdot\|$, there exists a sequence of matrices $B_1 \dots B_{2n-1}$ where $B_i$ is a circulant matrix if $i$ is odd, and a diagonal matrix otherwise, such that $\|B_1 B_2 \dots B_{2n-1} - A\| < \epsilon$, and where $S = circ(0, 1, 0, \dots, 0)$*

Because of their interesting properties, several researchers have considered circulant matrices as a replacement from full weight matrices inside neural networks.

## 3.2 THEORETICAL PROPERTIES OF DEEP DIAGONAL-CIRCULANT RELU NETWORKS

There has already been some recent theoretical work on Deep diagonal-circulant ReLU networks, in which 2-layer networks of unbounded width where shown to be universal approximators. These results are of limited interest, because the networks used in practice are of bounded width. Unfortunately, nothing is known about the theoretical properties of Deep diagonal-circulant ReLU networks in this case. In particular, the following questions remained unanswered up to now: Are Deep diagonal-circulant ReLU networks with bounded width universal approximators? What kind of functions can Deep diagonal-circulant ReLU networks with bounded-width and small depth approximate?

In this section, we first define formally *diagonal-circulant ReLU networks*, and then provide a theoretical analysis of their approximation capabilities.

**Definition 1** (Deep ReLU networks). *With $ReLU(x) = max(0, x)$, let $f_{A,b}(x) = ReLU\left(Ax + b\right)$ for any matrices $A \in \mathbb{C}^{n \times n}$, and any $b \in \mathbb{R}^n$. A* Deep ReLU network *is a function $f_{A_l, b_l} \circ \ldots \circ f_{A_1, b_1}$, where $A_1 \ldots A_l$ are arbitrary $n \times n$ matrices and $b_1 \ldots b_l \in \mathbb{C}^n$ and where $l$ and $n$ are the depth and the width of the network respectively.*

As in Moczulski et al. (2015), Deep diagonal-circulant ReLU networks can be defined as follows:

**Definition 2** (Deep diagonal-circulant ReLU networks). *A* Deep diagonal-circulant ReLU network *is a function $f_{D_l C_l, b_l} \circ \ldots \circ f_{D_1 C_1, b_1}$ where $D_1 \ldots D_l \in \mathbb{C}^{n \times n}$ are diagonal matrices, $C_1 \ldots C_l \in \mathbb{C}^{n \times n}$ are circulant matrices, and where $l$ and $n$ are the depth and the width of the network respectively.*

To show that bounded-width Deep diagonal-circulant ReLU networks are universal approximators, we first need a proposition relating standard deep neural networks to Deep diagonal-circulant ReLU networks.

**Proposition 2.** *Let $\mathcal{N} : \mathbb{R}^n \to \mathbb{R}^n$ be a deep ReLU networks of width $n$ and depth $l$, and let $\mathcal{X} \subset \mathbb{R}^n$ be a compact set. For any $\epsilon > 0$, there exists a deep diagonal-circulant ReLU network $\mathcal{N}'$ of width $n$ and of depth $(2n - 1)l$ such that $\|\mathcal{N}(x) - \mathcal{N}'(x)\|_2 < \epsilon$ for all $x \in \mathcal{X}$.*

We can now state the universal approximation corrolary:

**Corrolary 1.** *Bounded depth Deep diagonal-circulant ReLU networks are universal approximators on any compact set $\mathcal{X}$.*

*Proof.* Proposition 2 shows that bounded-width Deep diagonal-circulant ReLU networks can approximate any Deep ReLU network. It has been shown recently in Hanin (2017) that bounded-width deep ReLU networks are universal approximators. Together, these two results concludes the proof. □

It is important to remark that Deep diagonal-circulant ReLU networks are not necessarily more compact than Deep ReLU networks. Indeed, consider a $n$-wide Deep ReLU network with $l$ layers having $ln^2$ weights. The previous corollary tells us that this network can be decomposed in a Deep ReLU networks involving $l(2n-1)$ matrices, i.e. $2ln(2n - 1)$ weights.

Despite the lack of theoretical guarantees a number of work provided empirical evidence that bounded width and small depth Deep diagonal-circulant ReLU networks result in good performance (e.g. Moczulski et al. (2015); Araujo et al. (2018); Cheng et al. (2015)). The following theorem studies the approximation properties of these small depth networks.

**Proposition 3.** *Let $\mathcal{N} : f_{A_l, b_l} \circ \ldots \circ f_{A_1, b_1}$ be a deep ReLU network of width $n$ and depth $l$, such that each matrix $A_i$ is of rank $k_i$, where $k_i$ divides $n$. Let $\mathcal{X} \subset \mathbb{R}^n$ be a compact set. For any $\epsilon > 0$, there exists a deep diagonal-circulant ReLU network $\mathcal{N}'$ of width $n$ and of depth $\left(\sum_{i=1}^{n} \left(4k_i + 1\right)\right) l$ such that $\|\mathcal{N}(x) - \mathcal{N}'(x)\|_2 < \epsilon$ for all $x \in \mathcal{X}$.*

This result generalizes Proposition 2, showing that a Deep diagonal-circulant ReLU networks of bounded width and small depth can approximate a deep ReLU network in which the dense matrices are of low rank. Note in the proposition, we require that $k_i$ divides $n$. We conjecture that the proposition holds even without this condition, but we were not able to prove it.

Finally, what if we choose to use small depth network to approximate deep ReLU networks where matrices are not of low rank ? To answer this question, we first need to show the negative impact of replacing matrices by their low rank approximators in neural networks:

**Proposition 4.** *Let $\mathcal{N} = f_{A_l,b_l} \circ \ldots \circ f_{A_1,b_1}$ be a deep ReLU network, where $A_i \in \mathbb{C}^{n \times n}, b_i \in \mathbb{C}^n$ for all $i \in [l]$. Let $\bar{A}_i$ be the matrix obtained by a SVD approximation of rank $k$ of matrix $A_i$. Let $\sigma_{i,j}$ is the $j^{th}$ singular value of $A_i$. Define $\bar{\mathcal{N}} = f_{\bar{A}_l,b_l} \circ \ldots \circ f_{\bar{A}_1,b_1}$. Then, for any $x \in \mathbb{C}^n$, we have $\left\| \mathcal{N}(x) - \bar{\mathcal{N}}(x) \right\| \leq \frac{\left( \sigma_{max,1}^l - 1 \right) R \sigma_{max,k}}{\sigma_{max,1} - 1}$ where $R$ is an upper bound on norm of the output of any layer in $\mathcal{N}$, and $\sigma_{max,j} = \max_i \sigma_{i,j}$.*

Basically, this proposition shows that we can approximate matrices in a neural network by low rank matrices, and control the approximation error. In general, the term $\sigma_{max,1}^l$ could seem large, but in practice, it is likely that most singular values in deep neural network are small in order to avoid divergent behaviors. We can now prove the result on Deep diagonal-circulant ReLU networks:

**Corrolary 2.** *Consider any deep ReLU network $\mathcal{N} = f_{A_l,b_l} \circ \ldots \circ f_{A_1,b_1}$ of depth $l$ and width $n$. Let $\sigma_{max,j} = \max_i \sigma_{i,j}$ where $\sigma_{i,j}$ is the $j^{th}$ singular value of $A_i$. Let $\mathcal{X} \subset \mathbb{R}^n$ be a compact set. For any $k$ dividing $n$, there exists a deep diagonal-circulant ReLU network $\mathcal{N}' = f_{D_m C_m, b'_l} \circ \ldots \circ f_{D_1 C_1, b'_1}$ of width $n$ and of depth $m = 4(k+1)n$, such that for any $x \in \mathcal{X}, \|\mathcal{N}(x) - \mathcal{N}'(x)\| < \frac{\left( \sigma_{max,1}^l - 1 \right) R \sigma_{max,k}}{\sigma_{max,1} - 1}$, where $R$ is an upper bound on the norm of the outputs of each layer in $\mathcal{N}$.*

*Proof.* Let $\bar{\mathcal{N}} = f_{\bar{A}_l,b_l} \circ \ldots \circ f_{\bar{A}_1,b_1}$, where each $\bar{A}_i$ is the matrix obtained by a SVD approximation of rank $k$ of matrix $A_i$.

With proposition 4, we have an error bound on $\left\| \mathcal{N}(x) - \bar{\mathcal{N}}(x) \right\|$. Now each matrix $\bar{A}_i$ can be replaced by a product of $k$ diagonal-circulant matrices. By lemma 1, this product yields a Deep diagonal-circulant ReLU networks of depth $m = 4(k+1)n$, strictly equivalent to $\bar{\mathcal{N}}$ on $\mathcal{X}$. The result follows. □

## 4 EMPIRICAL EVALUATION

The experiments that we present in this section aim at answering the following questions. First question: what is the impact of increasing the number of diagonal-circulant factors on the accuracy of the network? To answer this question, we conduct a series of experiments on a synthetic classification dataset with an increasing number of factors. As we will show, the results match our theoretical analysis from Section 3. Second question: can this approach be useful to build more compact models in the context of large scale real-world machine learning applications. To answer this second question, we build a deep diagonal-circulant neural network architecture for video classification. The architecture is based on state-of-the-art architecture initially proposed by Abu-El-Haija et al. (2016b) and later improved by Miech et al. (2017) in involve several large layers that can be made more compact using circulant matrices as done in Araujo et al. (2018). As we will show, the approach demonstrate good accuracy and can be used to build a more compact network than the original one.

## 4.1 Impact of the number of diagonal-circulant factors on accuracy

**Experimental setup** The dataset is generated using the *make classification*[2] function from Scikit-Learn (Pedregosa et al., 2011). It is made of 10000 examples, 5 variables, 2 classes and 2 clusters for each class. We train a neural network with 3 hidden layers of 1024 neurons each. We used a batch size of 50, a learning rate of $5 \times 10^{-2}$, a learning rate decay of 0.9 every 10 000 examples. We compare the dense neural network with a Deep diagonal-circulant ReLU networks with several factors. We use the initialization proposed in Moczulski et al. (2015).

| #Factors | #Params | Compress. Rate (%) | Loss |
|----------|---------|--------------------|------|
| **Dense** | **2 107 397** | - | **0.16016** |
| $k = 2$ | 19 461 | 99.0 | 0.38847 |
| $k = 4$ | 35 845 | 98.2 | 0.36668 |
| $k = 8$ | 68 613 | 96.7 | 0.33275 |
| $k = 16$ | 134 149 | 93.6 | 0.32798 |
| $k = 32$ | 523 269 | 75.1 | 0.32657 |

Table 1: This table shows the loss obtain on the synthetic dataset given the number of factor used for each layer.

**Results** Table 4.1 shows the loss of the dense architecture versus the Deep diagonal-circulant ReLU networks with different factors. The table also shows the compression rate obtain with the Deep diagonal-circulant ReLU networks. We notice that the Deep diagonal-circulant ReLU networks manage to achieve more than $90\%$ compression rate with a substantial loss in accuracy with factor up to 16. Adding factors improve the accuracy but make the convergence difficult. We were note able to train a model with more than 32 layers. A solution would be to use the circulant-diagonal ReLU decomposition only on certain layer in order to trade-off compression with accuracy more precisely.

## 4.2 Deep diagonal-circulant ReLU networks for large-scale video classification

In this section, we demonstrate the applicability of diagonal-circulant ReLU networks in the context of a large scale video classification architecture trained on the Youtube-8M dataset. Our architecture is based on a state-of-the-art architecture that was initially proposed by Abu-El-Haija et al. (2016b) and later improved by Miech et al. (2017).

### 4.2.1 Experimental Settings

**Dataset** The dataset is composed of an embedding (each video and audio frames are represented by a vector of respectively 1024 and 128) of video and audio frames extracted every 1 seconds with up to 300 frames per video.

**Model Architecture** This architecture can be decomposed into three blocks of layers, as illustrated in Figure 4.1. The first block of layers, composed of the Deep Bag-of-Frames embedding, is meant to make an embedding of these frames in order to make a simple representation of each video. The first block of layers, composed of the Deep Bag-of-Frames embedding, is meant to process audio and video frames

---

[2]http://scikit-learn.org/stable/modules/generated/sklearn.datasets.make_classification.html

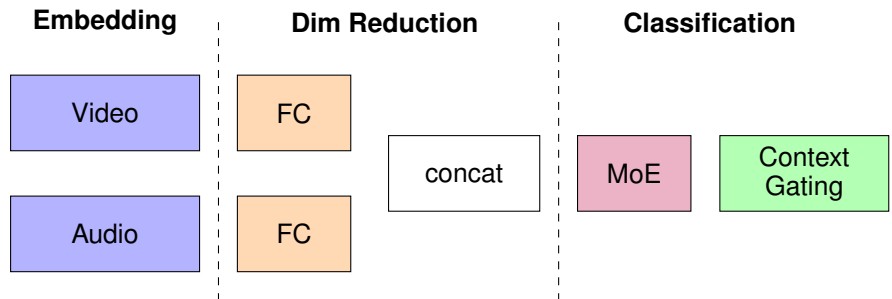

Figure 4.1: This figure shows the architecture used for the training of the YouTube-8M dataset.

independently. The DBoF layer computes two embeddings: one for the audio and one for the video. In the next paragraph, we will only focus on the describing the video embedding. (The audio embedding is computed in a very similar way.) A second block of layers reduces the dimensionality of the output of the embedding and merges the resulting output with a concatenation operation. Finally, the classification block uses a combination of Mixtures-of-Experts (MoE) Jordan & Jacobs (1993); Abu-El-Haija et al. (2016a) and Context Gating Miech et al. (2017) to calculate the final probabilities.

**Experiment**   We want to compare the effect on the circulant-diagonal ReLU decomposition only on certain layer to evaluate the trade-off between compression rate and accuracy. First, we train the architecture presented in Figure 4.1 without any circulant matrices to serve as a baseline. Then, we used the circulant-diagonal decomposition on each layer independently.

**Hyper-parameters**   All our experiments are developed with TensorFlow Framework Abadi et al. (2015). We trained our models with the CrossEntropy loss and used Adam optimizer with a 0.0002 learning rate and a 0.8 exponential decay every 4 million examples. We used a fully connected layer of size 8192 for the video DBoF and 4096 for the audio. The fully connected layers used for dimensionality reduction have a size of 512 neurons. We used 4 mixtures for the MoE Layer.

**Evaluation Metric**   We used the GAP (Global Average Precision), as used in Abu-El-Haija et al. (2016b), to compare our experiments.

### 4.2.2   RESULTS

This series of experiments aims at understanding the effect of circulant-diagonal ReLU decomposition over different layers with 1 factors. Table 2 shows the result in terms of number of weights, size of the model (MB) and GAP. We also compute the compression ratio with respect to the dense model. The compact fully connected layer achieves a compression rate of 9.5 while having a very similar performance, whereas the compact DBoF and MoE achieve a higher compression rate at the expense of accuracy. Figure 4.2 shows that the model with a compact FC converges faster than the dense model. The model with a compact DBoF shows a big variance over the validation GAP which can be associated with a difficulty to train. The model with a compact MoE is more stable but at the expense of its performance.

| Baseline Model | #Weights | Size (MB) | Compress. Rate (%) | GAP@20 | Diff. |
|---|---|---|---|---|---|
| Dense Model | 45 359 764 | 173 | - | **0.846** | - |
| Compact DBoF | 36 987 540 | 141 | 18.4 | 0.838 | -0.008 |
| Compact FC | 41 181 844 | 157 | 9.2 | 0.845 | **-0.001** |
| Compact MoE | 12 668 504 | 48 | 72.0 | 0.805 | -0.041 |

Table 2: This table shows the effect of circulant-diagonal decomposition on different layers.

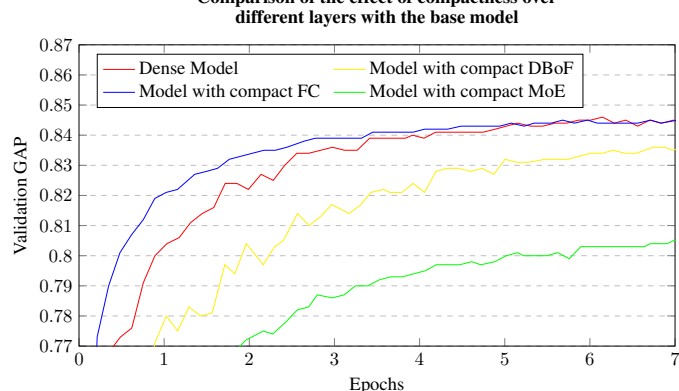

Figure 4.2: Validation GAP according to the number of epochs for different compact models.

## 5 CONCLUSIONS

In this paper we provided a theoretical study of the properties of Deep diagonal-circulant ReLU networks and demonstrated that they are bounded width universal approximators. The bound on this decomposition allowed us to calculate the error bound on any Deep diagonal-circulant ReLU networks given the depth on the network and the singular values associated with the weight matrices. Our empirical study demonstrate that we can trade-off model size for accuracy in accordance with the theory, and that we can use Deep diagonal-circulant ReLU networks in large scale machine learning applications.

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

## 6 SUPPLEMENTAL MATERIAL: TECHNICAL LEMMAS AND PROOFS

**Lemma 1.** *Let $A_l, \ldots A_1 \in \mathbb{C}^{n \times n}$, $b \in \mathbb{C}^n$ and let $\mathcal{X} \subset \mathbb{R}^n$ be a compact set. There exists $\beta_l \ldots \beta_1 \in \mathbb{C}^n$ such that for all $x \in \mathcal{X}$ we have $f_{A_l, \beta_l} \circ \ldots \circ f_{A_1, \beta_1}(x) = ReLU(A_l A_{l-1} \ldots A_1 x + b)$.*

*Proof.* of lemma 1. Define $\Omega = \max_{x \in \mathcal{X}, j \in [l]} \left\| \prod_{k=1}^{j} A_k x \right\|_{\infty}$. Define $h_j(x) = A_j x + \beta_j$. Let $\beta_1 = \Omega \mathbf{1}_n$ where $\mathbf{1}_n$ is the $n$-vector of ones. Clearly, for all $x \in \mathcal{X}$ we have $h_1(x) \geq 0$, so $ReLU \circ h_1(x) = h_1(x)$. More generally, for all $j < n - 1$ define $\beta_{j+1} = \mathbf{1}_n \Omega - A_{j+1} \beta_j$. It is easy to see that for all $j < n$ we have $h_j \circ \ldots \circ h_1(x) = A_j A_{j-1} \ldots A_1 x + \mathbf{1}_n \Omega$. This garantees that for all $j < n$, $h_j \circ \ldots \circ h_1(x) = ReLU \circ h_j \circ \ldots \circ ReLU \circ h_1(x)$. Finally, define $\beta_l = b - A_l \beta_{l-1}$. We have, $ReLU \circ h_l \circ \ldots \circ ReLU \circ h_1(x) = ReLU(A_j \ldots A_1 x + b)$. □

*Proof.* of proposition 2. Assume $\mathcal{N} = f_{A_l, b_l} \circ \ldots \circ f_{A_1, b_1}$. By theorem 1, for any $\epsilon' > 0$, any matrix $A_i$, there exists a sequence of $2n - 1$ matrices $C_{i,n} D_{i,n-1} C_{i,n-1} \ldots D_{i,1} C_{i,1}$ such that $\left| \prod_{j=0}^{n-1} D_{i,n-j} C_{i,n-j} - A_i \right| < \epsilon'$, where $D_{i,1}$ is the identity matrix. By lemma 1, we know that there exists $\{\beta_{ij}\}_{i \in [l], j \in [n]}$ such that for all $i \in [l]$, $f_{D_{in} C_{in}, \beta_{in}} \circ \ldots \circ f_{D_{i1} C_{i1}, \beta_{i1}}(x) = ReLU(D_{in} C_{in} \ldots C_{i1} x + b_i)$.

Now if $\epsilon'$ tends to zero, $\| f_{D_{in} C_{in}, \beta_{in}} \circ \ldots \circ f_{D_{i1} C_{i1}, \beta_{i1}} - ReLU(A_i x + b_i) \|$ will also tend to zero for any $x \in \mathcal{X}$, because the ReLU function is continuous and $\mathcal{X}$ is compact. Let $\mathcal{N}' = f_{D_{1n} C_{1n}, \beta_{1n}} \circ \ldots \circ f_{D_{i1} C_{i1}, \beta_{i1}}$. Again, because all functions are continuous, for all $x \in \mathcal{X}$, $\| \mathcal{N}(x) - \mathcal{N}'(x) \|$ tends to zero as $\epsilon'$ tends to zero. □

**Proposition 5.** *Let $A \in \mathbb{C}^{n \times n}$ a matrix of rank $k$. Assume that $n$ can be divided by $k$. For any $\epsilon > 0$, there exists a sequence of $4k + 1$ matrices $B_1, \ldots, B_{4k+1}$, where $B_i$ is a circulant matrix if $i$ is odd, and a diagonal matrix otherwise, such that*

$$\left\| A - \prod_{i=1}^{4k+1} B_i \right\|_F < \epsilon$$

*Proof.* of proposition 5. Let $U\Sigma V^T$ be the SVD decomposition of $M$ where $U, V$ and $\Sigma$ are $n \times n$ matrices. Because $M$ is of rank $k$, the last $n-k$ columns of $U$ and $V$ are null. In the following, we will first decompose $U$ into a product of matrices $WRO$, where $R$ and $O$ are respectively circulant and diagonal matrices, and $W$ is a matrix which will be further decomposed into a product of diagonal and circulant matrices. Then, we will apply the same decomposition technique to $V$. Ultimately, we will get a product of $4k + 2$ matrices alternatively diagonal and circulant.

Let $R = circ(r_1 \ldots r_n)$. Let $O$ be a $n \times n$ diagonal matrix where $O_{i,i} = 1$ if $i \le k$ and $0$ otherwise. The $k$ first columns of the product $RO$ will be equal to that of $R$, and the $n-k$ last colomns of $RO$ will be zeros. For example, if $k = 2$, we have:

$$RO = \begin{pmatrix} r_1 & r_n & 0 & \cdots & 0 \\ r_2 & r_1 & & & \\ r_3 & r_2 & \vdots & & \vdots \\ \vdots & \vdots & & & \\ r_n & r_{n-1} & 0 & \cdots & 0 \end{pmatrix}$$

Let us define $k$ diagonal matrices $D_i = diag(d_{i1} \ldots d_{in})$ for $i \in [k]$. For now, the values of $d_{ij}$ are unknown, but we will show how to compute them. Let $W = \sum_{i=1}^{k} D_i S^{i-1}$. Note that the $n-k$ last columns of the product $WRO$ will be zeros. For example, with $k = 2$, we have:

$$W = \begin{bmatrix} d_{1,1} & & & & d_{2,1} \\ d_{2,2} & d_{1,2} & & & \\ & d_{2,3} & \ddots & & \\ & & & \ddots & \\ & & & d_{2,n} & d_{1,n} \end{bmatrix}$$

$$WRO = \begin{pmatrix} r_1 d_{11} + r_n d_{21} & r_n d_{11} + r_{n-1} d_{21} & 0 & \cdots & 0 \\ r_2 d_{12} + r_1 d_{22} & r_1 d_{12} + r_n d_{22} & & & \\ & & & \vdots & & \vdots \\ \vdots & \vdots & & & \\ r_n d_{1n} + r_{n-1} d_{2n} & r_{n-1} d_{1n} + r_{n-2} d_{2n} & 0 & \cdots & 0 \end{pmatrix}$$

We want to find the values of $d_{ij}$ such that $WRO = U$. We can formulate this as linear equation system. In case $k = 2$, we get:

$$\begin{pmatrix} r_n & r_1 & & & & & \\ r_{n-1} & r_n & & & & & \\ & & r_1 & r_2 & & & \\ & & r_n & r_1 & & & \\ & & & & r_2 & r_3 & \\ & & & & r_1 & r_2 & \\ & & & & & & \ddots \\ & & & & & & & \ddots \end{pmatrix} \times \begin{pmatrix} d_{2,1} \\ d_{1,1} \\ d_{2,2} \\ d_{1,2} \\ d_{2,3} \\ d_{1,3} \\ \vdots \\ \vdots \end{pmatrix} = \begin{pmatrix} U_{1,1} \\ U_{1,2} \\ U_{2,1} \\ U_{2,2} \\ \\ \\ \vdots \end{pmatrix}$$

The $i^{th}$ bloc of the bloc-diagonal matrix is a Toeplitz matrix induced by a subsequence of length $k$ of $(r_1, \ldots r_n, r_1 \ldots r_n)$. Set $r_j = 1$ for all $j \in \{k, 2k, 3k, \ldots n\}$ and set $r_j = 0$ for all other values of $j$. Then it is easy to see that each bloc is a permutation of the identity matrix. Thus, all blocs are invertible. This entails that the block diagonal matrix above is also invertible. So by solving this set of linear equations, we find $d_{1,1} \ldots d_{k,n}$ such that $WRO = U$. We can apply the same idea to factorize $V = W'.R.O$ for some matrix $W'$. Finally, we get

$$A = U\Sigma V^T = WRO\Sigma O^T R^T W'^T$$

Thanks to Theorem 1, $W$ and $W'$ can both be factorized in a product of $2k - 1$ circulant and diagonal matrices. Note that $O\Sigma O^T$ is diagonal, because all three are diagonal. Overall, $A$ can be represented with a product of $4k + 2$ matrices, alternatively diagonal and circulant. $\qquad \square$

*Proof.* of proposition 3 By proposition 5, each low rank matrix of the neural net can be decomposed in a small number of diagonal and circulant matrices. By lemma 1, the matrices can be connected to form a neural net. $\qquad \square$

*Proof.* of proposition 4 Let $x_0 \in \mathbb{C}^n$ and $\bar{x}_0 = x_0$. For all $i \in [l]$, define $x_i = ReLU(A_i x_{i-1} + b)$ and $\bar{x}_i = ReLU(\bar{A}_i \bar{x}_{i-1} + b)$. By lemma 2, we have

$$\|x_i - \bar{x}_i\| \le \sigma_{i,k+1} \|x_{i-1}\| + \sigma_{i,1} \|x_{i-1} - \bar{x}_{i-1}\|$$

Observe that for any sequence $a_0, a_1 \ldots$ defined reccurently by $a_0 = 0$ and $a_i = ra_{i-1} + s$, the reccurence relation can be unfold as follows: $a_i = \frac{s(r^i - 1)}{r - 1}$. We can apply this formula to bound our error as follows

$$\|x_l - \bar{x}_l\| \le \frac{(\sigma_{max,1}^l - 1)\sigma_{max,k} \max_i \|x_i\|}{\sigma_{max,1} - 1}. \qquad \square$$

**Lemma 2.** *Let $A \in \mathbb{C}^{n \times n}$ with singular values $\sigma_1 \ldots \sigma_n$, and let $x, \bar{x} \in \mathbb{C}^n$. Let $\bar{A}$ be the matrix obtained by a SVD approximation of rank $k$ of matrix $A$. Then we have:*

$$\left\| ReLU(Ax + b) - ReLU(\bar{A}\bar{x} + b) \right\| \le \sigma_{k+1} \|x\| + \sigma_1 \|\bar{x} - x\|$$

*Proof.* Recall that $\|A\|_2 = \sup_z \frac{\|Az\|_2}{\|z\|_2} = \sigma_1 = \|\bar{A}\|_2$, because $\sigma_1$ is the greatest singular value of both $A$ and $\bar{A}$. Also, note that $\|A - \bar{A}\|_2 = \sigma_{k+1}$. Let us bound the formula without ReLUs:

$$\begin{aligned}
\left\|(Ax+b) - \left(\bar{A}\bar{x}+b\right)\right\| &= \left\|(Ax+b) - \left(\bar{A}\bar{x}+b\right)\right\| \\
&= \left\|Ax - \bar{A}x - \bar{A}\left(\bar{x}-x\right)\right\| \\
&\leq \left\|\left(A-\bar{A}\right)x\right\| + \left\|\bar{A}\right\|_2 \left\|\bar{x}-x\right\| \\
&\leq \|x\| \sigma_{k+1} + \sigma_1 \left\|\bar{x}-x\right\|
\end{aligned}$$

Finally, it is easy to see that for any pair of vectors $a, b \in \mathbb{C}^n$, we have $\|ReLU(a) - ReLU(b)\| \leq \|a-b\|$. This concludes the proof. $\qquad\square$

