# OpenReview forum: "The Expressive Power of Deep Neural Networks with Circulant Matrices"
_ICLR.cc/2019/Conference_

### Official Review · AnonReviewer1 · 2018-11-02
**neither original nor thorough enough [title no longer appropriate after rebuttals]**

**Rating:** 6
**Confidence:** 5

**Review:**

The paper proposes using structured matrices, specifically circulant and diagonal matrices, to speed up computation and reduce memory requirements in NNs. The idea has been previously explored by a number of papers, as described in the introduction and related work.  The main contribution of the paper is to do some theoretical analysis, which is interesting but of uncertain impact.

The experiments compare performance against DeepBagOf`Fframes (DBOF) and MixturesOfExperts (MOE). However, there are other algorithms that are both more competitive and more closely related. I would like to see head-to-head comparisons with tensor-based algorithms such as Novikov et al: https://papers.nips.cc/paper/5787-tensorizing-neural-networks, which achieves huge compression ratios (~200 000x), and other linear-algebra based approaches.

AFTER READING REBUTTAL
I've increased my score because the authors point out previous work comparing their decomposition and tensortrains (although note the comparisons in Moczulski are on different networks and thus hard to interpret) and make a reasonable case that their work contributes to improve understanding of why circulant networks are effective.

I strongly agree with authors when they state: "We also believe that this paper brings results with a larger scope than the specific problem of designing compact neural networks. Circulant matrices deserve a particular attention in deep learning because of their strong ties with convolutions: a circulant matrix operator is equivalent to the convolution operator with circular paddings".  I would broaden the topic to structured linear algebra more generally. I hope to someday see a comprehensive investigation of the topic.

---

> ### Author Response · Authors · 2018-11-13
> **About the impact of our contribution**
>
>
> We would like to thank both reviewers for a valuable feedback and we apologize for the typos and grammatical errors. We have double checked the current version so hopefully there won't be anymore in the next revision.
>
> You (Reviewer 2) have expressed some doubts about the significance of our theoretical contributions. We would like to clarify and emphasize some points hereafter.
>
> Firstly a comparison between the decomposition we use and other decompositions such as TensorTrain has already been published in ICLR [2]. As one can see in Table 1 and Figure 4 from [2], the circulant matrix decomposition compares favorably to TensorTrain. For completeness, we will add a comparison between TensorTrain and our decomposition on the YouTube dataset (in Table 2) but this should not change the essence of our contribution.
>
> Secondly, our main contribution is of importance: we provide a similar approximation result as [3] did for classical networks, but for circulant networks. Indeed we provide a bound on the approximation error of a circulant neural network with bounded width and height. We believe that this result can be of interest to anyone trying to build compact networks using circulant matrices.
>
> More importantly, without the result we provide in our paper,  the good results reported in [2,1] were difficult to explain with the existing theory: [9] states that any linear operator can be decomposed into a product of at least n diagonal and circulant factors (where n can be as big as 1024), but in practice good results have been observed in [1,2] with as few as 1 factor. So in a sense, the situation is analogous to the one of  neural networks based on fully connected layers *before* the first (celebrated) results on approximation with bounded nets [3,4].
>
> We also believe that this paper brings results with a larger scope than the specific problem of designing compact neural networks. Circulant martices deserve a particular attention in deep learning because of their strong ties with convolutions: a circulant matrix operator is equivalent to the convolution operator with circular paddings (as shown in [5]). This fact makes any contribution to the area of circulant matrices particularly relevant to the field of deep learning with impacts beyond the problem of designing compact models.
> For instance, it is currently not known whether convolutional neural networks are universal approximators. Our work proves that a particular type of convolutional neural nets are universal approximators. We believe that this is a strong first result that paves the way to more general results about error bounds  in general CNNs.
>
> Finally, regarding the architecture, we choose the Deep Bag-of-Frames (DBoF) and Mixtures of Experts (MoE) architectures since they are  state of the art in the computer vision area, as discussed in [6, 7, 8].
>
>
> [1] Cheng, Yu, et al. "An exploration of parameter redundancy in deep networks with circulant projections." Proceedings of the IEEE International Conference on Computer Vision. 2015.
>
> [2] Moczulski, Marcin, et al. "ACDC: A structured efficient linear layer." ICLR (2016).
>
> [3] Barron, A. R. (1993). "Universal approximation bounds for superpositions of a sigmoidal function. IEEE Transactions on Information theory, 39(3), 930-945.
>
> [4] Hanin, Boris. "Universal function approximation by deep neural nets with bounded width and relu activations." arXiv preprint arXiv:1708.02691 (2017).
>
> [5] Xiao et al. "Dynamical Isometry and a Mean Field Theory of CNNs: How to Train 10,000-Layer Vanilla Convolutional Neural Networks." ICML 2018
>
> [6] Abu-El-Haija et al. "YouTube-8M: A Large-Scale Video Classification Benchmark", arXiv preprint arXiv:1609.08675
>
> [7] Miech et al. "Learnable pooling with Context Gating for video classification", Proc. of the CVPR Workshop on YouTube-8M Large-Scale Video Understanding (2017)
>
> [8] Paul Natsev, "Context-Gated DBoF Models for YouTube-8M", https://static.googleusercontent.com/media/research.google.com/fr//youtube8m/workshop2018/natsev.pdf
>
> [9]  Huhtanen, M., & Perämäki, A. (2015). Factoring matrices into the product of circulant and diagonal matrices. Journal of Fourier Analysis and Applications, 21(5), 1018-1033.

---

### Official Review · AnonReviewer2 · 2018-11-03
**An important contribution.**

**Rating:** 7
**Confidence:** 4

**Review:**

In this paper, the authors prove that bounded width diagonal-circulant ReLU networks (I will call them DC-ReLU henceforth) are universal approximators (this was shown previously without the bounded width condition). They also show that bounded width and small depth DC-ReLUs can approximate deep ReLU nets with row rank parameters matrices. This explains the observed success of such networks. The authors also provide experiments to demonstrate the compression one can achieve without sacrificing accuracy.

Pros: The authors provide strong approximation results that explain the observed success of DC-ReLUs.

Cons: Too many grammatical errors (mainly improper pluralization of verbs and punctuation errors), typos, stylistic inconsistencies seriously affect the readability of the paper. The authors should pay more attention to these.

---

### Official Review · AnonReviewer3 · 2018-11-13
**Claims not sufficiently justified**

**Rating:** 4
**Confidence:** 4

**Review:**

The experiments in the paper are similar to those explored in previous work! The main contribution claimed in the paper is the theoretical formulation for compact design of neural networks using circulant matrices instead of fully connected matrices.

I do not think the claim is sufficiently justified by the theoretical results provided.

Earlier result already shows how any matrix fully connected matrices can be approximated by 2n-1 circulant matrices. As the authors themselves point out, this theoretical result does not necessarily imply reduction in number of parameters since the for a depth l network, the equivalent diagonal-circulant-ReLU network will now require (2n-1)l depth, or 2n(2n-1)l parameters.

The main results (Proposition 3, 4) show that if the fully connected networks of depth l network are parameterized by (approximately) rank k matrices, then the resultant depth of diagonal-circulant network required to approximate the original network is (4k+1)l, which results in a total of 8n(4k+1)l parameters. Similar to the case of full rank fully connected networks (proposition 2), this result does not necessarily indicate a compression of number of parameters either. In particular, if fully connected networks are indeed rank k, then we only need nkl parameters parameters to represent the matrix, which is lower than the number of parameters required by the diagonal-circulant network.

So I do not see how the result can be seen as a justification for using diagonal-circulant networks as compact representations.

Writing:
Theorem 1: The statement about approximability with B_1B_2…B_{2n-1} is independent of p and S.
Proposition 3: The expression for depth should be \sum_{i=1}^l (4k_i+1)  — sum should go from i=1 to l and there should be no multiplicative factor l

Other non-critical comments: Multiplication by circulant matrices amounts to circular convolution with full dimensional kernel. In this sense, replacing a fully connected layers by circulant matrices is similar to replacing it with convolutional layers.  May be this connection can be explicitly stated in the paper.

---

> ### Author Response · Authors · 2018-11-16
> **Expressivity of circulant matrices**
>
> Thanks for the review, this is an interesting comment and will address it in the next revision of the paper.
>
> The circulant diagonal decomposition is in fact more general than the low rank matrix factorization in the sense that any low rank matrix can be represented using a circulant matrix, but converse is not necessarily true.
> It is true that the circulant diagonal generally requires more parameters, however it only requires linearly more parameters. This explain the circulant diagonal decomposition generally performs better in practice as demonstrated in [1], (table 1).
>
> Another important difference  is the computational complexity of the matrix-vector multiplication. With low rank decomposition the matrix vector multiplication O(nk^2) wheras with a circulant diagonal decomposition it is O(k n log(n))
>
>
> [1] Moczulski, Marcin, et al. "ACDC: A structured efficient linear layer." ICLR (2016).

---

> ### Author Response · Authors · 2018-11-21
> **justification for using diagonal-circulant networks as compact representations.**
>
> We would like to answer your comment "I do not see how the result can be seen as a justification for using diagonal-circulant networks as compact representations" in more precise terms.
>
> There are plenty of matrix decomposition methods aiming at compact representation. For e.g., Givens-matrix decompositions, DCT-diagonal products, product of Toeplitz matrices, Diagonal-Hadamard products, low rank decomposition, etc..
>
> Up to know, there was no theoretical argument supporting the choice of one of these compact representations.
>
> We argue that the most basic requirement for a compact representation is that it must allow to compactly represent low-rank matrices. Products of Circulant-Diagonal matrices do satisfy this requirement. It is unknown for the others decompositions, and we conjecture that it does not hold for Toeplitz.
>
> Moreover, we argue that circulant-diagonal products are MORE EXPRESSIVE than low-rank decomposition, because:
> - Any rank-k matrix (represented by 2nk parameters) can be represented by a circulant-diagonal products involving O(nk) parameters
> - The converse is not true: there exists full-rank circulant matrices, so these circulant matrices (represented by n parameters) are rank-n matrices requiring n^2 parameters

---

### Meta-Review · Area_Chair1 · 2018-12-13

**Confidence:** 5
**Recommendation:** Reject

**Metareview:**

The paper conveys interesting study but the reviewers expressed concerns regarding the difference of this work compared to existing approaches and pointed a room for more thorough empirical evaluation.